# Label-Free Cholesteric Liquid Crystal Biosensing Chips for Heme Oxygenase-1 Detection within Cerebrospinal Fluid as an Effective Outcome Indicator for Spontaneous Subarachnoid Hemorrhage

**DOI:** 10.3390/bios12040204

**Published:** 2022-03-29

**Authors:** Hui-Tzung Luh, Yi-Wei Chung, Po-Yi Cho, Yu-Cheng Hsiao

**Affiliations:** 1Department of Neurosurgery, Taipei Medical University-Shuang Ho Hospital, New Taipei 23561, Taiwan; 17260@s.tmu.edu.tw; 2Taipei Neuroscience Institute, Taipei Medical University, New Taipei 23561, Taiwan; 3Department of Neurosurgery, School of Medicine, College of Medicine, Taipei Medical University, Taipei 11031, Taiwan; 4Graduate Institute of Clinical Medicine, National Taiwan University, Taipei 10617, Taiwan; 17256@s.tmu.edu.tw; 5Stanford Byers Center for Biodesign, Stanford, CA 94305, USA; 6Division of Cardiology, Department of Internal Medicine, Taipei Medical University-Shuang Ho Hospital, New Taipei 23561, Taiwan; 7Taipei Heart Institute, Taipei Medical University, Taipei 11031, Taiwan; 8Cardiovascular Research Center, Taipei Medical University Hospital, Taipei 11031, Taiwan; 9Division of Cardiology, Department of Internal Medicine, School of Medicine, College of Medicine, Taipei Medical University, Taipei 11031, Taiwan; 10Graduate Institute of Biomedical Optomechatronics, College of Biomedical Engineering, Taipei Medical University, Taipei 11031, Taiwan; m850108002@tmu.edu.tw; 11Cell Physiology and Molecular Image Research Center, Wan Fang Hospital, Taipei Medical University, Taipei 11696, Taiwan

**Keywords:** cholesteric liquid crystal, heme oxygenase-1, cerebrospinal fluid, spontaneous subarachnoid hemorrhage

## Abstract

A novel device for cholesteric liquid crystal (LC; CLC)-based biosensing chips for detecting heme oxygenase (HO)-1 within the cerebrospinal fluid (CSF) was invented. In the CLC device, the reorientation of the LCs was strongly influenced by the alignment layer surface and adjacent LCs. When the substrate was coated with the alignment layer, the CLCs oriented homeotropically in a focal conic state. Once HO-1 was immobilized onto the orientation sheet-coated substrate, the CLC changed from a focal conic state to a bright planar state by disrupting the CLCs. The concentration of HO-1 within CSF was shown to be an effective outcome indicator for patients with a spontaneous subarachnoid hemorrhage. We showed that the CLC immunoassaying can be used to measure HO-1 with a lower detection limit of about 10 ng/mL. The linear range was 10 ng/mL to 1 mg/mL. An easy-to-use, rapid-detection, and label-free CLC immunoassay device is proposed.

## 1. Introduction

Spontaneous subarachnoid hemorrhage (SAH) is a catastrophic condition that is commonly linked with severe neurological impairment and significant morbidity and mortality [1,2,3]. In theory, after the occurrence of the SAH, the hemolysis of the red blood cells in subarachnoid space leads to rapid hemoglobin releasing which is metabolized to heme and then iron and bilirubin [4,5,6]. Among the stages, heme oxygenase (HO)-1, an inducible isozyme of HO, is the rate-limiting enzyme in the degradation of hemoglobin [7,8]. Several molecules involved in posthemorrhagic heme metabolism were implicated in the etiology of early brain damage and delayed vasospasms following SAH in previous experimental studies [9,10,11,12,13].

Building on these ideas, multiple clinical investigations have found links between different biochemical components of the cerebrospinal fluid (CSF) associated with heme metabolic pathways and various neurological outcomes in patients with SAH [14,15,16,17,18]. According to a study conducted by Wang et al., HO-1 in CSF was found to be a poor outcome predictor in patients with Fisher grade III aneurysmal SAH [19,20,21]. However, the lack of a low-cost and rapid HO-1 test technique has hampered its practical adoption.

Recently, label-free and sensitive liquid crystal (LC) biosensors have been developed successfully. The immune binding reaction of LC molecules reorientates their direction and alters their light signal. Changes in LC optical properties enable visual detection in label-free immunoassays [22]. The re-directed LCs are reported to be sensitive to changes in immunobinding responses and light signals from devices [23,24]. Furthermore, the previous study combined the LC with a microfluidic device to detect ethanol and bovine serum albumin (BSA) [25,26]. In addition to nematic LC, cholesteric LC (CLC) has unique optical properties such as flexibility, bistability, and Bragg reflection [27,28,29]. The Bragg reflection property of CLC enables it to be seen as color [29]. The first CLC sensor device used for biological detection was created in 2015 [30] and a highly sensitive colorful CLC biosensor has since been invented. However, the manufacture of CLC biosensors requires complex fabrication processes that must be confined to defined areas, such as TEM grids [27] or a cell device [31]. (In addition, CLC biosensors can be integrated with smartphones, so various disease biomarkers can be detected at home.) In order to simplify the fabrication process, a single-substrate CLC device was invented.

In this paper, we show a new CLC-based biosensor chip for HO-1 detection in CSF. The behavior between paired HO-1 antigen/antibody and CLC molecules was first investigated. This paper proposes that this CLC-based biosensing chip substantially differs from a typical biosensor. The HO-1 antigens/antibodies can be measured by the optical characteristics of the CLC biosensor below the cross-polarized microscopy. A delicate interface between the CLC molecules and the arranged layer of N,N-dimethyl-n-octadecyl-3-aminopropyltrimethoxysilyl chloride (DMOAP) was used to measure the concentrations of HO-1. Because DMOAP has a strong attraction, it can attract biomolecules to the surface. A schematic illustration of the CLC biosensor for HO-1 is displayed in Figure 1.

## 2. Materials and Methods

### 2.1. Patients

The Taipei Medical University Joint Institutional Review Board (TMU-JIRB #N201805074) authorized this study which was carried out in compliance with human ethical rules. The research enrolled patients aged 20~90 years who had spontaneous SAH and were receiving external ventricular drainage (EVD) only, EVD combined with an endovascular intervention, or EVD combined with a surgical craniotomy. Patients were excluded if they or their families declined additional medical treatment after being recruited. Written informed consent was acquired from the patient or the patient’s next of kin in the case of unconscious patients.

### 2.2. CSF Sample Collection, Preparation, and Analysis

Following a SAH diagnosis, intraventricular CSF was acquired through EVD the same day, the second day, and the third day. In our treatment approach, EVD was generally put in place within 24 h of a patient being diagnosed with spontaneous SAH in the emergency room, and the EVD was left in place for 10~14 days. CSF samples were immediately centrifuged at 900× *g* for 20 min at 4 °C. Within 30 min, the supernatant was collected and frozen at −80 °C. A Human HO-1 enzyme-linked immunosorbent assay (ELISA) Kit (ADI-EKS-800, Enzo Life Sciences^®^, Farmingdale, NY, USA) was used to measure CSF HO-1 concentrations.

### 2.3. Study Design

Patients recruited in this study were receiving routine care from a multidisciplinary team that included neurosurgeons, neurointensivists, and interventional neuroradiologists. The management approach included resuscitation, early surgical or endovascular obliteration of the aneurysm, routine intracranial pressure and neurointensive care management, and aggressive medical or endovascular therapy for vasospasms if present. Patients were monitored in the neurointensive care unit following surgery.

The Glasgow Outcome Scale-Extended (GOS-E) and modified Rankin Scale (mRS) were used to assess functional outcomes at 30, 90, and 180 days after onset. Patients were grouped as having a favorable outcome (mRS scores of 1~3 or GOSE scores of 5~8) or an unfavorable outcome (mRS scores of 4~6 or GOS-E scores of 1~4) [20,21].

### 2.4. CLC Biosensor Preparation

The CLC material used in this study used nematic LC E7 (Merck, Darmstadt, Germany) as a host and R5011 (Merck) as a chiral dopant. The concentration of chiral dopant R5011 was ~2.6 wt%, and the major reflection of CLCs is located close to 590 nm. The glass substrate was immersed in a 1% DMOAP aqueous solution at room temperature for 15 min to coat the DMOAP in a vertically aligned layer on the glass substrate and was rinsed with deionized (DI) water for 1 min to remove excess DMOAP solution on the glass surface. In addition, an anti-HO-1 antibody was immobilized with 1 ng/mL onto the DMOAP-coated glass substrate in an aqueous solution. After rinsing with DI water for 1 min to remove the excess anti-HO-1 aqueous solution on the substrate, the HO-1 concentrations of 1 mg/mL, 1 μg/mL, and 1 ng/mL of the cerebrospinal fluid from the patients was immobilized on the coated glass. To fabricate a CLC sandwiched cell, silicon ball spacers mixed with ethanol were distributed on the DMOAP-coated slide, which was covered with another DMOAP-coated glass. Finally, the empty CLC sandwiched cell was filled with CLC material by the capillary action method to form a CLC biosensor. The cell gap thickness of the complete CLC biosensor was about 10 μm.

## 3. Results and Discussion

### 3.1. Detecting HO-1 by the CLC Biosensor

The design of the CLC biosensor is shown in Figure 1. Immobilization of the anti-HO-1 antibody and HO-1 by the DMOAP-coated glass is illustrated in Figure 1. Finally, the device was filled with CLCs. In the absence of HO-1, CLC are aligned vertically near the substrates by DMOAP and the CLC is the FC/P mix-state, resulting in light scattering. When the HO-1 is adsorbed on the DMOAP-coated substrate, the vertical anchoring power of the CLCs is weakened, making CLC transfer to the total P structure without the light scattering property. A polarized optical image under cross-polarization and the optical mechanism of the CLC biosensor are also shown in Figure 2. Optical textures of two CLC biosensors with different concentrations of HO-1 are displayed in Figure 2. Furthermore, we can observe that the optical brightness of the optical texture increases with rising HO-1 concentrations. The random focal conic (FC) states are also shown in non-biological molecules close to two substrates. When the HO-1 is immobilized in the coated substrate, the CLC biosensor is completely in the planar (P) state. Light passing through the CLC material is scattered by part of the FC state of the CLC layer. However, the vertical anchoring power of the aligned-DMOAP layers was reduced by immobilizing the HO-1. With the increase in HO-1 concentration, the CLC molecules switched from the FC to the P state. Finally, the perfect P-state change to a higher optical intensity of the CLC biosensor is shown in Figure 2. 

In order to realize the quantitative research of the CLC biosensor, the spectra of the CLC biosensor device immobilized with different concentrations of HO-1 are demonstrated in Figure 3. The transmittance of the reflection band of the CLC biosensor was augmented with an increasing HO-1 concentration. When the HO-1 concentration increases, the P-state of CLC dominates in the device. Therefore, the optical response of CLC arises due to Bragg reflection. The photonic band is more complete with increasing the HO-1 concentration. We determined the detection limit of the CLC biosensor to be 10 ng/mL of HO-1. Based on these properties, a log-scale CLC biosensor as a positive correlation behavior is also proposed in Figure 4. The drawing of the calibration curve comes from the minimum value of the light intensity of the photon band for quantification. The sample size for the error bar was 10. The linear dependence between the transmittance of the Bragg reflection of CLC and different HO-1 concentrations was measured (Figure 4). These results demonstrate that the Bragg reflection of CLC in the spectrum can be used to detect and quantify the HO-1 in a linear manner.

### 3.2. Clinical Study

In clinical cases, we could identify generally low concentrations of HO-1 in CSF from patients suffering from spontaneous SAH on the initial day (day 0) and the next day (day 1) of ictus. However, from day 2, variations in HO-1 concentrations among patients occurred (Figure 5). In addition, HO-1 concentrations in CSF on day 2 were related to patients’ functional outcomes 6 months after the ictus of SAH. Functional outcomes of patients were measured by mRS (0~6) (Figure 6a) and GOS-E (1~8) (Figure 6b), and we found that the higher the HO-1 concentration was, the worse was the functional outcome.

HO-1 is an important enzyme for rapid heme metabolism and protection against oxidative damage both in vivo and in vitro. Matz and colleagues showed that marked induction of HO-1 messenger (m)RNA was detected at 6 and 24 h after a lysed blood injection in a SAH rat model induced by direct blood injection into the cisterna magna [26]. Considering the metabolic timeline of hemoglobin degradation and synthesis of the HO-1 protein, we assumed that significant variations in HO-1 concentrations on day 2 of the SAH ictus represented different oxidative stress levels in the subsequent disease course.

In clinical practice, the long-term prognoses of patients suffering from spontaneous SAH greatly influence the medical decisions that their families and attending physicians make. A low-cost, quick, and reliable detection technology could efficiently provide more information to achieve more precise medical decisions. Wang et al. confirmed that HO-1 in CSF 1 week after ictus is a poor outcome predictor in patients with Fisher grade III aneurysmal SAH [19]. In our cohort, we further identified that HO-1 concentration in CSF on day 2 of ictus of SAH could serve as a poor outcome predictor. We have not only shown the predictive value of the HO-1 concentration in CSF as early as 2 days after ictus, but we have also developed CLC technology to provide a low-cost, quick, and reliable detection method.

### 3.3. Detecting HO-1 by the CLC Biosensors from CSF

We used the CSF from recruited spontaneous SAH patients to replace the standard HO-1 sample. After preparing CLC samples containing CSF, we measured the light intensity by spectrophotometry. The light information-dependent differences in spontaneous SAH patients were measured. Transmitted intensities of Bragg’s reflection of the CLC biosensor with time in different patients suffering from spontaneous SAH are shown in Figure 7. The detection time was about half an hour for immobilization. The brightness of the device rose when the HO-1 concentration increased under spectrophotometry. From the experimental data, HO-1 concentrations in CSF samples of patients significantly increased after the second day, resulting in a substantial increase in the light intensity. This result was the same as the ELISA experimental results. Results of the CLC biosensor showed that the density of HO-1 was positively correlated with the light transmittance. Some of the CSF samples whose concentration was too low were not clearly measured by the ELISA. They could be effectively judged by the CLC system, and it was observed that the HO-1 concentration rose on the first day by the CLC system. Compared with ELISA, this study proves that CLCs have huge potential for development as a cheap, sensitive, and rapid biosensing technique for SAH patients to measure HO-1. Despite the potential of this CLC biosensor, there are some limitations such as a difficult fabrication process and the use of a spectrometer. However, the limitations can be solved by using LCD production and AI image recognition technology.

## 4. Conclusions

A CLC biosensing chip for detecting HO-1 in CSF was proposed. The reorientation of the CLCs was strongly influenced by the alignment layer surface and adjacent CLCs. When HO-1 was immobilized onto the DMOAP-coated substrate, the CLC changed from a focal conic state to a bright planar state by disrupting the CLCs. The concentration of HO-1 within CSF was proposed as an effective outcome indicator for patients with spontaneous SAH. We successfully proved that CLCs can measure HO-1 concentrations in patients with spontaneous SAH. The HO-1 concentration in CSF samples of patients significantly increased after the second day. The linear range was 10 ng/mL~1 mg/mL for the CLC biosensor to measure standard HO-1 concentrations. An innovative, easy-to-use, label-free, and rapid-detection CLC device for HO-1 detection is proposed.

## Figures and Tables

**Figure 1 biosensors-12-00204-f001:**
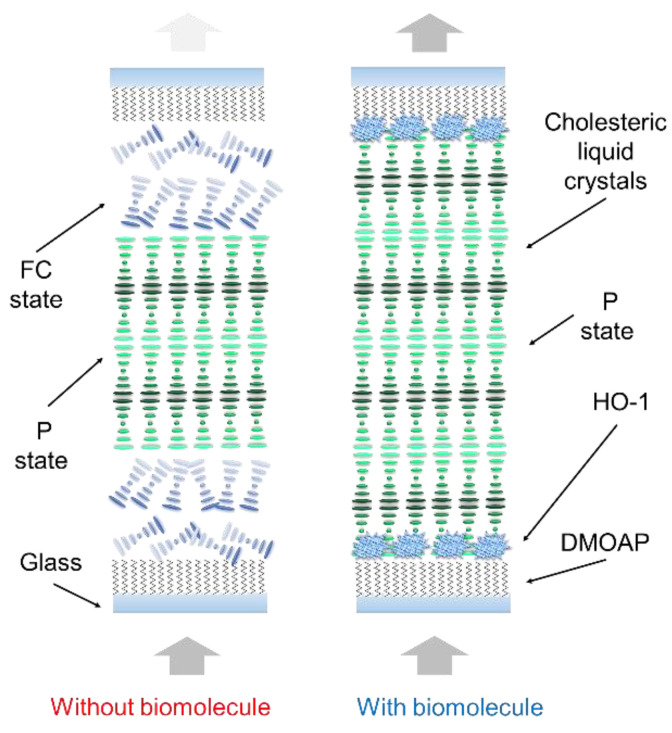
Illustration of the cholesteric liquid crystal (CLC) biological sensor chip in the presence of heme oxygenase (HO)-1 biomolecules in the DMOAP-coated cell. Different color arrows indicate the different light penetration intensities.

**Figure 2 biosensors-12-00204-f002:**
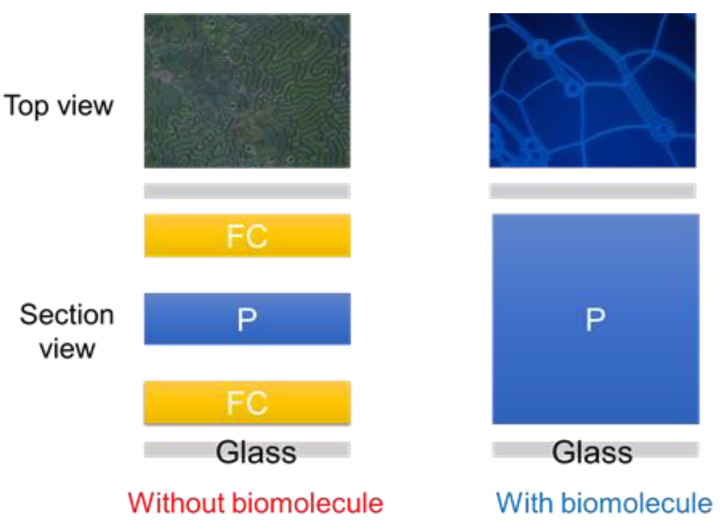
Optical mechanism and structure design of the cholesteric liquid crystal (CLC) biosensor both with and without the presence of biomolecules.

**Figure 3 biosensors-12-00204-f003:**
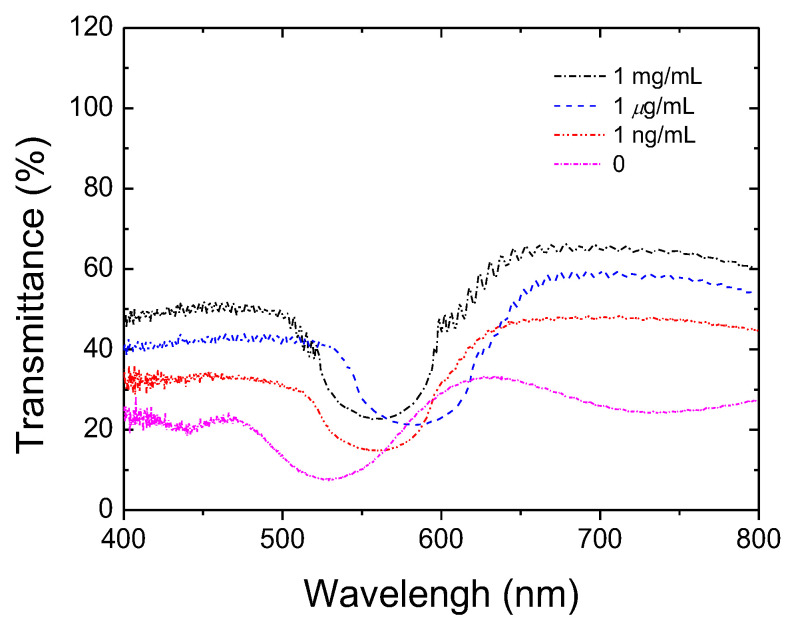
Spectra of the cholesteric liquid crystal (CLC) biological sensor immobilized with various concentrations of heme oxygenase (HO)-1 (0~1 mg/mL).

**Figure 4 biosensors-12-00204-f004:**
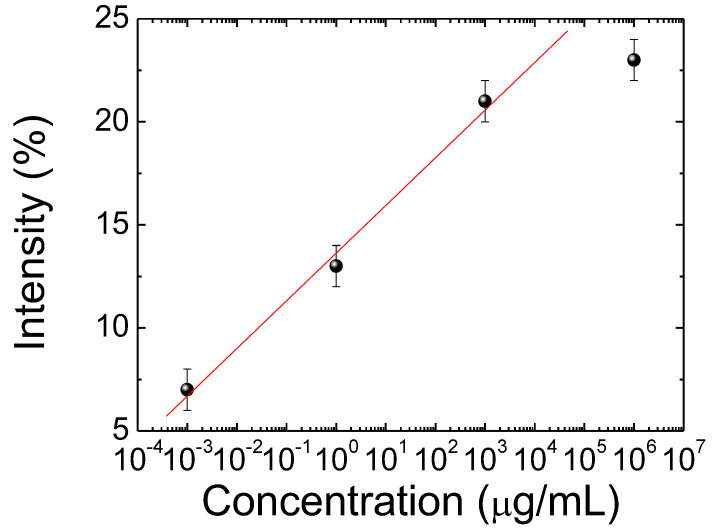
Linear positive correlations of the minimum transmittance of the photonic band of the cholesteric liquid crystal (CLC) biological sensor at different heme oxygenase (HO)-1 concentrations.

**Figure 5 biosensors-12-00204-f005:**
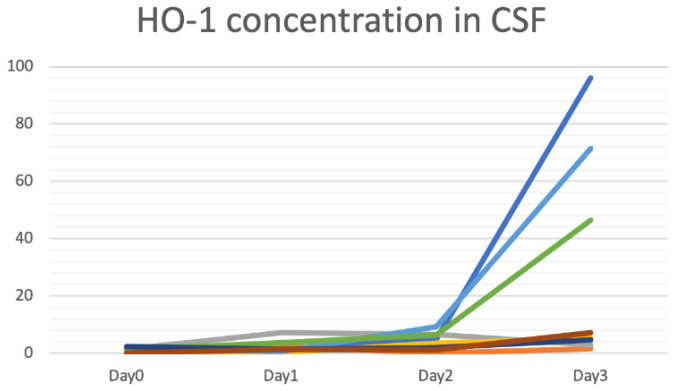
Trends of heme oxygenase (HO)-1 concentration in cerebrospinal fluid (CSF) from recruited spontaneous subarachnoid hemorrhage (SAH) patients. (*x*-axis: days from ictus; *y*-axis: HO-1 concentration in CSF (ng/mL)). (Data from ELISA).

**Figure 6 biosensors-12-00204-f006:**
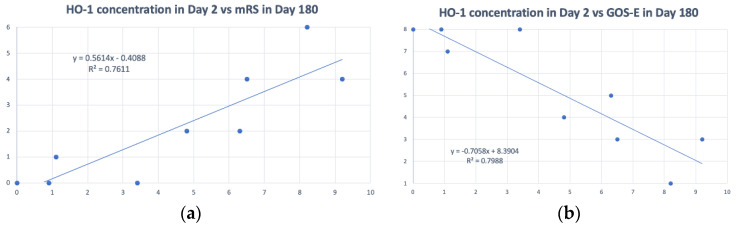
(**a**) Heme oxygenase (HO)-1 concentration on day 2 correlated with the modified Rankin Scale (mRS) on day 180. (*x*-axis: HO-1 concentration in cerebrospinal fluid (CSF) (ng/mL) *y*-axis: mRS from 0 to 6) (**b**) HO-1 concentration on day 2 correlated with the Glasgow Outcome Scale-Extended (GOS-E) on day 180. (*x*-axis: HO-1 concentration in CSF (ng/mL); *y*-axis: GOS-E from 1 to 8). (Data from ELISA).

**Figure 7 biosensors-12-00204-f007:**
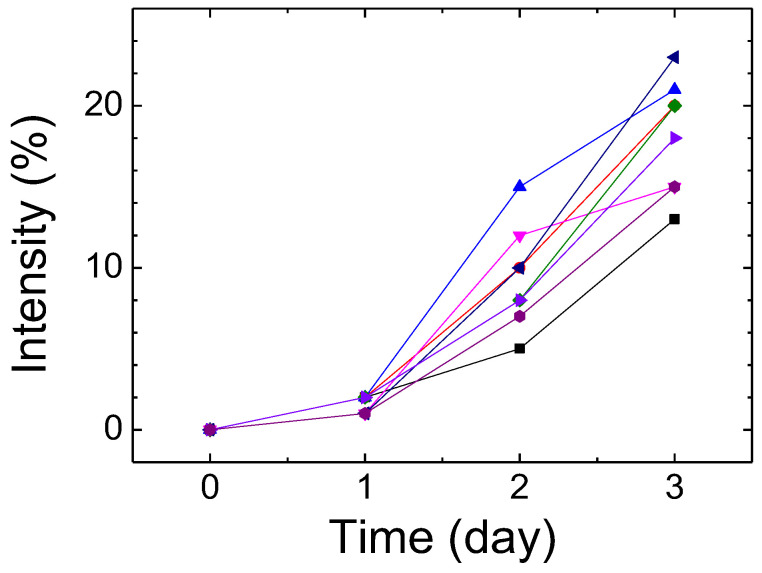
The transmittance intensity of Bragg’s reflection of the cholesteric liquid crystal (CLC) biosensor with time in different patients suffering from spontaneous subarachnoid hemorrhage (SAH).

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
