# Peer review of "Label-Free Cholesteric Liquid Crystal Biosensing Chips for Heme Oxygenase-1 Detection within Cerebrospinal Fluid as an Effective Outcome Indicator for Spontaneous Subarachnoid Hemorrhage"

_biosensors, 2022, doi:10.3390/bios12040204_

Round 1

Reviewer 1 Report

The manuscript presents a liquid crystal (LC)-based biosensor for heme oxygenanse-1 detection in cerebrospinal fluid (CSF). Though the study is conceptually novel, the results are not well presented. Therefore, I think the current article is not suitable for publication in the journal like Biosensors. Before resubmitting or submitting to other journal, authors need to pay attention to the following points:

  1. The authors should summarize the current techniques for sensing HO-1 and their limitations and how their method improves these limitations.
  2. The authors should elaborate the CLC-based sensing principle in more details for the readers and should clarify how this method is better than other methods.
  3. The authors should explain in more detail what is DMOAP and why is it needed to immobilize the anti-HO-1.
  4. Figure 3 shows the transmittance spectrum of the CLC biosensor with HO-1 concentrations 1000 times apart (0, 1ng/mL, 1ug/mL, and 1mg/mL). But the authors should demonstrate more concentrations in the middle (like 10x apart).
  5. Figure 3 shows that with increased concentration of HO-1, the peak of the transmittance spectrum also shifts right (with an exception to 1mg/mL). The reviewer is curious if the authors can use this as a potential sensing mechanism for lower concentration ranges ( 0-1ug/mL).
  6. Also, to prove the effectiveness of the sensor, the authors must show selectivity results i.e. the sensor does not respond to analytes other than HO-1.
  7. The authors should clarify how the calibration curve in Figure 4 is plotted from Figure 3. The y-axis value in Figure 3 does not look to match the transmittance values of Figure 3. Please clarify. Also, the author must specify the sample size for the error bar in figure 4.
  8. The authors reported two different LODs—10 ng/mL in the abstract and 1 fg/mL in section 3.1. The authors need to clarify this.

Reviewer 2 Report

Dear Authors, 

Congratulations on your well-written manuscript. These are some feedback you may consider, to make the manuscript better.

  1. Some of the affiliations are similar. Not sure if they can be combined?
  2. Remove the extra space at the start of no. 11 in affiliation.
  3. In line 28, maybe you can change the word from “proved” to “showed”.
  4. In line 53, maybe you can change the word from “said” to “reported”.
  5. In line 55, maybe you can state what is the detection limit of ethanol and the BSA from references 14 and 15.
  6. In line 57, do elaborate on the advantage of Bragg reflection.
  7. In line 71, you have referred to Figure 1. However the Figure was inserted into the text much later. Consider moving Figure 1 near line 71.
  8. In the biosensor preparation section, do insert the name of manufacturer for the different materials used.
  9. I am sure many of our readers will be curious about how the cell will actually look. Is it possible to insert an actual picture of the cell / sensor near line 113?
  10. Consider revising line 125 on the different states.
  11. In Figure 1 label what do the different color of the arrows refer to.
  12. In figure 2, label all the parts and describe right how the picture insert shown in figure 2 was taken.
  13. In Figure 4 and 7, consider stating which specific wavelength was used in the plot.
  14. In Figures 5 and 6, state which sensor was used to measure the concentration.
  15. In the conclusion, Line 220, you stated that the linear range was up to 1 mg. However from Figure 4, the linear response was not. Please kindly elaborate. 

Thank you and good luck! Looking forward to your next development and manuscript.

Reviewer 3 Report

This manuscript presented a device for cholesteric liquid crystal-based biosensing chips for detecting heme oxygenase (HO)-1 within cerebrospinal fluid (CSF). The mechanism was fabricated based on the random focal conic (FC) states were also shown in non-biological molecules close to two substrates. When the HO-1 immobilized in the coated substrate, the CLC biosensor was completely in the planar (P) state. Light passing through the CLC material is scattered by part of the FC state of the CLC layer. The HO-1 antigens/antibodies could be measured by the optical characteristics of the CLC biosensor below the cross-polarized microscopy. It’s a interesting work. However, in order to improve the manuscript, I think the following points should be paid attention to:

  1. I doubt that the amount of anti-HO-1 antibody onto the DMOAP-coated glass substrate only 1 ng/mL, while the HO-1 concentrations were 1 mg/mL, 1 μg/mL, and 1 ng/mL.
  2. How the authors to calculate thedetection limit?
  3. It is confused that linear range in Abstract (10 ng/mL to 1 mg/mL) and Conclusions(1 pg/mL~1 mg/mL).
  4. In Abstract, the first and last sentences are repeated.
  5. Line 29, a full point is missed.
  6. Line 141, a full point is surplus.

Round 2

Reviewer 1 Report

I am still not convinced by the authors response on the selectivity. The author mentions that many published papers reported the selectivity of CLC biosensor. Can the authors cite those papers? Are those papers from the same design i.e. same material composition, same antibody-target pair? 

If not, it is important to test the selectivity for this work.  Moreover, this selectivity test can work as a control for verifying the results. Without the control, it is difficult to conclude that the change in transmittance in Figure 3 is from specific interaction of the antibody-target pair. 

Author Response

This manuscript is a resubmission of an earlier submission. The following is a list of the peer review reports and author responses from that submission.